# Metabarcoding in Diet Assessment of *Heterotrigona itama* Based on trnL Marker towards Domestication Program

**DOI:** 10.3390/insects12030205

**Published:** 2021-02-28

**Authors:** Jaapar Fahimee, Aqilah Sakinah Badrulisham, Mohd Sani Zulidzham, Nurul Farisa Reward, Nizar Muzammil, Rosliza Jajuli, Badrul Munir Md-Zain, Salmah Yaakop

**Affiliations:** 1Agrobiodiversity and Environmental Research Centre, MARDI Headquarter, 43400 Serdang, Selangor, Malaysia; miesre@mardi.gov.my (J.F.); zulidzham@mardi.gov.my (M.S.Z.); nurulfarisa@mardi.gov.my (N.F.R.); muzzammil@mardi.gov.my (N.M.); rosliza@mardi.gov.my (R.J.); 2Centre for Insect Systematics, Department of Biological Sciences and Biotechnology, Faculty of Science and Technology, Universiti Kebangsaan Malaysia, 43600 Bangi, Selangor, Malaysia; aqilah.sakinah@yahoo.com; 3Department of Biological Sciences and Biotechnology, Faculty of Science and Technology, Universiti Kebangsaan Malaysia, 43600 Bangi, Selangor, Malaysia; abgbadd@ukm.edu.my

**Keywords:** stingless bee, DNA, plant species, food source, meliponiculture, domestication, genetic, captivity

## Abstract

**Simple Summary:**

*Heterotrigona itama* is the most abundant stingless bee species commonly found in many ecosystems and is actively domesticated in the meliponiculture industry. In this research, the honey quality is the main criterion to be evaluated and is correlated with the plant species that the bees consume as their main diet. The objective of this study was to investigate the diet of *H. itama* derived from 12 populations in Malaysia. As a result, five plant phyla and 70 plant families with 262 species were obtained, of which four species were the abundant species consumed by *H. itama*. These findings are very valuable for strategizing the management of the *H. itama* domestication process specifically in a mono-cropping system and in a netted structure.

**Abstract:**

Honey quality is the main criterion used for evaluating honey production in the stingless bee *Heterotrigona itama*, and it is correlated with the plant species consumed as its main diet. The objective of this study was to obtain the metabarcode data from 12 populations of *H. itama* species throughout Malaysia (Borneo and Peninsular Malaysia) using the trnL marker. A total of 262 species under 70 families and five phyla of plants were foraged by *H. itama* in the studied populations. Spermatophyta and Magnoliophyta were recorded as the two most abundant phyla foraged, at 55.95% and 32.39%, respectively. Four species, *Garcinia oblongifolia, Muntingia calabura, Mallotus pellatus,* and *Pinus squamata*, occurred abundantly and were consumed by *H. itama* in all the populations. These data are considered as a fundamental finding that is specific to the diet of *H. itama* for strategizing the management of the domestication process specifically in a mono-cropping system and in a netted structure. Thus, based on these findings, we recommend *Momordica charantia, Melastoma* sp., and *Cucumis sativa* as the best choices of food plant species to be planted and utilized by *H. itama* in meliponiculture.

## 1. Introduction

The stingless bee is known as a pollinator of many plant species in forests and agricultural ecosystems. For example, stingless bee species have become a pollinating agent of the mangosteen (*Garcinia mangostana*), liposu (*Bacaurea lanceolate*), and star fruit (*Averrhoa bilimbi*) [1]. From over 500 species of stingless bees recorded worldwide [2], about 45 species are found in Malaysia [3], with *Heterotrigona itama* Cockerell, 1918 being the most common and abundant species in most ecosystems. This is most likely due to the availability of its food sources [4]. The ubiquitous *H. itama* has become the domesticated species of choice in our local meliponiculture industry [5], as well as in neighboring countries [6]. To date, 50,000 colonies of *H. itama* have been reared for honey production in Malaysia and this trend is increasing yearly. 

According to Vit et al. (2013) [7], *H. itama* is able to continue its honey production due to the exclusive interactions and coevolution between the stingless bees and specific angiosperm species of flowering trees [8]. In the meliponiculture industry, the breeders and the market players are keen to evaluate the honey quality, which is determined by the plant nectar and/or pollens that are consumed by the bees. In this regard, the best flowering plant species in relation to *H. itama* should be planted as the main food source, for in situ reproduction, as well as for ex situ conservation [9].

According to Afik et al. (2014) [10], Absy et al. (2018) [11], and Jaapar et al. (2019) [12], *H. itama* normally collects pollen, nectar, and resin when it forages away from the nest. However, in some cases, the bees also collect and obtain nutrients or salts from animal carcasses [13]. Therefore, in order to utilize this species within the setting of a netting structure, it is crucial that the food plant species is prioritized. This is because the plants depend solely on the stingless bees to act as their pollinating agent to perform pollination and, simultaneously, for the operators to increase the honey yield [14]. Furthermore, the stingless bee rearing program within the orchard, and in the natural habitat, was determined as the best practice to support the ecosystem services and to protect the bee species from population decline and extinction [15].

The application of DNA barcoding has become the latest approach or tool in biotechnology, showing benefits for human health and marine sustainability [16]. The application of DNA molecules has also solved many problems and issues related to the meliponiculture industry worldwide, such as accessing the honey compounds in several honey varieties [17], stingless bee identification through DNA barcoding [18], and plant species identification as food sources for the stingless bees [19].

Information on the plant species foraged or used as the diet for *H. itama* has never been widely investigated despite the great success of *H. itama* in the meliponiculture industry, specifically in honey production. However, a lot of information is available on the pollen collection of *H. itama*, although the question as to whether the bees consume pollen or only store it in pollen pots has yet to be clarified. In this regard, metabarcoding analysis by using a trnL marker on the *H. itama* diet is highly crucial and is the main objective for this study. It is important that the flowering plants consumed in the natural or wild habitat by *H. itama* are nurtured on the farms or orchards located together with *H. itama* colonies for sustainable reproduction; this will also ensure the best quality of honey production. 

## 2. Materials and Methods

### 2.1. Sampling of the Stingless Bees

A total of 36 individuals of *Heterotrigona itama* were sampled and collected from 12 meliponiculture sites throughout Malaysia (Table 1 and Figure 1), with three individuals collected from three colonies from each locality.

### 2.2. DNA Extraction

Three individuals from different colonies in each locality were dissected and the DNA was extracted from the whole body using NucleoSpin^®^ DNA Insect protocols (Macherey-Nagel, Germany). DNA samples from three individuals (25 µL × 3) were pooled from each locality prior to amplification and totaling up to only 12 DNA samples for the amplification process. 

### 2.3. DNA Sample QC

The quality of the pooled purified DNAs samples was monitored on 1% TAE agarose gel. The DNA concentration was measured using a spectrophotometer (Implen NanoPhotometer^®^ N60/N50) and fluorometric quantification using an iQuant™ Broad Range dsDNA Quantification kit. Preparation of the library was done through two rounds of polymerase chain reaction (PCR). The first PCR was carried out using trnL primers, PCR forward primer, g-A49425 (5′-GGGCAATCCTGAGCCAA-3′) and PCR reverse primers, H-B49466 (5′- CCATTGAGTCTCTGCACCTATC-3′) [20]. All the PCR reactions were carried out with Q5^®^ High-Fidelity DNA Polymerase (NEB). A total of 25 µL first PCR MasterMix consisted of 12.5 µL of 2× KAPA HiFi HotStart ReadyMix, 5 µL of forward and reverse primers, and 2.5 µL DNA. The amplification was performed under the following protocol: initial denaturation of 95 °C for 3 min, followed by 25 cycles of denaturation at 95 °C for 30 s, primer annealing at 55 °C for 30 s and extension at 72 °C for 30 s, with final elongation at 72 °C for 5 min, and stored at 4 °C until PCR clean-up. 

The DNA was purified using AMPure XP beads to purify the trnL amplicon away from free primers and primer dimers in the PCR clean-up procedure. The AMPure Xp beads were taken out and left at room temperature. Amplicon PCR products were centrifuged at 1000× *g* at 20 °C for 1 min to collect condensation. AMPure XP beads were vortexed for 30 s to make sure that the beads were evenly dispersed. A total of 20 µL of AMPure XP beads were added to each well of the amplicon PCR plate and entire volumes were pipetted up and down 10 times gently. Then the sample was incubated for 5 min at room temperature. The plate was placed on a magnetic stand for 2 min and then the supernatant was removed. Then, the beads were washed with 80% alcohol. Each sample was added to 200 µL 80% ethanol and was incubated for 30 s and the supernatant was removed afterwards. This procedure was repeated twice and ethanol was removed. The amplicon PCR plate was placed on the magnetic stand for 10 min to allow the beads to air-dry. Then, the amplicon PCR plate was removed from the magnetic stand and to each well was added 52.5 µL of 10 mM Tris pH 8.5 followed by mixing to ensure that the beads were fully resuspended. Amplicon PCR plates were incubated at room temperature for 2 min, then placed on the magnetic stand for 2 min until the supernatant was clear. Only 50 µL of supernatant from the amplicon PCR plate were transferred to a new 96-well PCR plate.

### 2.4. Second PCR (Index PCR)

This involved a dual indices Illumina sequencing adapter using a Nextera XT index kit. The PCR mixture consisted of 25 µL of 2× KAPA HiFi HotStart ReadyMix, 5 µL of Index 1 primers (N7XX) and Index 2 primers (S5XX), and 10 µL PCR grade water, then the plate was centrifuged at 1000× *g* at 20 °C for 1 min. PCR amplification for DNA templates with indexes was performed using the following profiles: polymerase activation at 95 °C for 3 min, followed by 8 cycles of denaturation at 95 °C for 30 s, annealing at 55 °C for 30 s, and extension at 72 °C for 30 s, and final extension at 72 °C for 5 min. The quality of the libraries was measured using Agilent Bioanalyzer 2100 System by Agilent DNA 1000 Kit and fluorometric quantification by Helixyte GreenTM Quantifying Reagent.

### 2.5. Next-Generation Sequencing

The libraries were normalized and pooled according to the protocol recommended by Illumina and proceed to sequencing using MiSeq platform using 250 PE.

### 2.6. Data Analysis

Paired-end reads were first had sequence adaptors and low-quality reads removed using BBDuk of the BBTools package. After this, the forward and reverse reads were merged using USEARCH v11.0.667 [21]. All sequences that were shorter than 150 bp or longer than 600 bp were removed from the downstream processing. Reads were then aligned with trnL sequences based on NCBI database. After these quality assessment steps, reads were clustered de novo into operational taxonomic units (OTUs) at 97% similarity using UPARSE v11.0.667 [22]; rare OTUs with fewer than 2 reads (doubleton), which were often spurious, were deleted from downstream processing. A single representative sequence from each OTU was randomly chosen, and Pynast was used to align and construct a phylogenetic tree to compare with NCBI database. Then, taxonomic assignment of OTUs was achieved using QIIME V1.9.1 [23] A phylogenetic dendrogram was constructed at the species level using Bray–Curtis distance with 1000 bootstraps to define the relationship between localities of *H. itama* using Paleontological Statistics Software Package for Education (PAST 3) software. Other statistical analysis was done in R V3.6.1 [24].

## 3. Results

A total of 527,398 plant trnL sequences were generated from 12 samples of *H. itama* and ranged from 7448 to 56,612. Sabah TE showed the most reads (70,068), followed by Perlis (56,612), Sarawak Beladin (55,714), Kedah (54,937), and Perak (52,216), while the fewest reads were from Pool Sarawak-Bintulu (7448). The Shannon–Wiener index (H’) indicated that Sabah TE had the highest diversity with H’ of 2.73 and 63 OTU, and the lowest was recorded for Selangor at 0.644 and 19 OTU. The same was found for Simpson 1-D and Chao1, in which Sabah TE showed the most OTUs (1 − D = 0.899, Chao1 = 83), while the fewest was for *H. itama* from Selangor (1 − D = 0.243, Chao1 = 22.33) (Table 2). Furthermore, the OTU number from Borneo (Sabah TE, Sabah Telipok, Sarawak Kuching, Pool Sarawak Bintulu, Sarawak Beladin, Sabah Tuaran, and Sarawak Miri) was 328 and higher than that of Peninsular Malaysia (159). The rarefaction curve was not asymptote since the methodology yielded up to 7500 reads only. *H. itama* from Sabah TE and Sabah Telipok showed the highest sequencing depth compared to the other localities (Figure 2).

A total of 262 species from 70 families and five phyla were assigned at 97% similarity (Figure 3). The most common family recorded was Euphorbiaceae (27), followed by Fabaceae (20), Cucurbitaceae (16) and Fagaceae (13). Spermatophyta was the most abundant phylum (55.95%) recorded in samples from all the study localities, followed by Magnoliophyta (32.39%), Tracheophyta (4.72%), and Anthophyta (1.97%), while the least was Embryophyta (0.13%). There were also 4.85% unknown phyla due to the blast not hitting any plant DNA (Table 3). The Sabah Telipok data showed the highest abundance of Spermatophyta (18.24%), followed by Perak (17.07%), Selangor (15.20%), and Kedah (11.76%), while Pool Sarawak Bintulu showed the lowest (0.34%). Furthermore, Embryophyta was only recorded from Sarawak Beladin, indicating that this locality represented all the bee food plant phyla in this study (Table 4). 

Based on the analyzed data, we found that samples from Sarawak Beladin presented all the six phyla that were collected by *H. itama*, while Magnoliophyta represented the most common phylum collected, at 78.7% (Table 3). Most of the Tracheophyta were found in Sabah Tuaran (18%), Anthophyta in Sabah Telipok (10.56%), Magnoliophyta in Sarawak Beladin (87.7%) Spermatophyta in Perak (95.8%), Embryophyta in Sarawak Beladin (1.16%), and the unknown phyla mostly found in Pool Sarawak Bintulu (80.8%) (Table 4). Most of the species in Tracheophyta were found in Sabah Tuaran (18%), Anthophyta in Sabah Telipok (10.56%), Magnoliophyta in Sarawak Beladin (87.7%) Spermatophyta in Perak (95.8%), and Embryophyta in Sarawak Beladin (1.16%), while an unknown phylum was mainly found in Pool Sarawak Bintulu (80.8%) (Table 4).

The heat map constructed showed the 30 most abundant plant species for 12 localities foraged by *H. itama* (Figure 4). The most dominant species found in all locations were *Garcinia oblongifolia, Muntingia calabura, Mallotus pellatus, Pinus squamata*, and *Phoebe puwenensis*. Meanwhile, *Umtiza listeriana, Spathodea campanulate, Ischaemum aristatum,* and *Dillenia indica* could only be found in several localities. Samples from Perak (P) and Perlis (Pr) showed that *H. itama* had consumed more from *Garcinia oblongifolia* (*p* = 56.71%, Pr = 69.63%), followed by *Muntingia calabura* (*p* = 32.67%, Pr = 21.34%).

Meanwhile, Sabah TE (STE), and Sabah Telipok (SBT) samples showed that the most abundant food plant utilized by *H. itama* was *Phoebe puwenensis* (STE = 25.82%, SBT = 31.56%). Kedah (K) and Selangor (S) showed the highest totals for *Phoenix canariensis* (K = 37.55%, S = 87.01%), while other samples showed that *H. itama* foraged mainly for *Macarangga tanarius* (Sarawak Beladin = 67.04%), *Garcinia mangostana* (Terengganu = 41.08%), *Annona reticulata* (Sabah Tuaran = 30.58%), *Cucumis melo* (Sarawak Miri = 49.93%), and *Pinus squamata* (Sarawak Bintulu = 4.89%). 

The lowest totals of plant DNA found from all localities were: *Typha angustifolia* (Sabah Telipok = 0.167), *Aclepias tuberosa* (Perak = 0.16), *Heliamphora pulchella* (Perlis = 0.004%), *Triptilion cordifolium* (Sabah TE = 0.15%), *Xanthocercis rabiensis* (Sarawak Beladin = 0.076%), *Chamaecrista polita* (Kedah = 0.006%), *Macarangga bancana* (Sarawak Kuching = 0.01%), *Iryanthera sagotiana* (Terengganu = 0.058%), *Mallotus mollisimus* (Sabah Tuaran = 0.071%), *Manihot pruinose* (Sarawak Miri = 0.003%), *Brassica oleraceae* (Selangor = 1.680%), and *Laguncularia racemose* (Pool Sarawak Bintulu = 0.582%) (Figure 5). In addition, there were overlaps in plant species between the sampling sites (Figure 6). Samples from Sabah and Sarawak showed a larger overlap compared to between Sarawak and Peninsular Malaysia. There were close relationships of plant communities between results from Terengganu and Sarawak, Miri, and Sarawak, Kuching, and Sabah, Tuaran based on the phylogenetic dendrogram constructed based on Bray–Curtis distance (Figure 7).

## 4. Discussion

The information on the plant species that were foraged by the stingless bee, *H. itama* has been summarized in this study based on the data obtained from metabarcoding analysis. The metabarcode has been presented and proven to yield more thorough and rigorous findings compared to the melissopalynology technique [25]. Chloroplast and nuclear barcoding could be amplified from the pollen, and therefore, the metabarcode would give the best and most rapid means of identifying the plant species. Selvaraju et al. (2019) [26] noted that there were only 60 species under 34 families of trees revealed by the melissapalynology analysis from the west coast Malaysia. Likewise, Ghazi (2015) [27] also found only 59 plant species in an island ecosystem, and most of the species came from the underutilized fruits. However, more than 140 species of pollen from the melissapalynology analysis remained unidentified [1], which indicated that the method is still lacking compared to metabarcoding. A total of 262 plant species under 70 families was recorded in this study, therefore, we concluded that a broad diversity of plant species had been foraged by *H. itama*. Furthermore, in terms of the total plant species detected, our totals were distinctly higher than the melissapalynology method. Interestingly, an understory level plant family, i.e., Vitaceae, and a top canopy level family, i.e., Bombaceae, have been recorded to be consumed and foraged by *H. itama* using the metabarcode analysis. In addition, the species *Pterisanthes stonei*, considered as rare in the secondary forest, had also been recorded in the diet of *H. itama,* which is strongly supported by the results of the metabarcode in this study.

The samples of *H. itama* were collected from various localities throughout Malaysia (Peninsular Malaysia and Borneo) to represent a holistic view of *H. itama*’s diet, in consideration of its limited range of foraging behavior of up to 300 m distances only [28], and at a specific time and for a specific duration [29]. The data which were obtained from 12 localities at different settings of the meliponiculture sites presented a novel finding for understanding the diet of *H. itama*. Flower pollens are basically collected by *H. itama,* and then transferred to its hind legs before being stored in the pollen pots [30]. This behavior highlighted that *H. itama* either consumed the pollen or unintentionally attached the pollen onto its body. Therefore, the metagenomic analysis using trnL can explore the plant species really selected by *H. itama* as food for its energy sources [31]. The results obtained can be used as a management tool to domesticate *H. itama* species in the netting structure for pollination purposes in honey production. The data can be applied at meliponicultural sites to enhance the quality of the honey [8].

Generally, all *H. itama* samples collected from Borneo (Sabah and Sarawak) showed a large number of OTUs, ranging from 35 to 64, as compared to Peninsular Malaysia (19–42) (Table 2). This data supported the species richness of the food plants, which is much higher on Borneo compared to Peninsular Malaysia, as similarly found in the study of [32]. Our results also showed that in Borneo, the highest diversity index was from Sabah as compared to Sarawak. In observing the high richness and diversity of plants in Borneo, Saw and Chung (2015) [33] and Pereira et al. (2019) [34] reported that over 12,000 taxa of seed plants occur in Sabah and Sarawak, while only 8200 species occur in Peninsular Malaysia. According to Wan Mohd Jaafar et al. (2020) [35], a large coverage of forests would support a high diversity of plants species, as represented in Terengganu, Perak, and Kedah, with 654,625 ha, 1,019,052 ha, and 330,585 ha of forest area, respectively, which correspondingly serve as the main habitats for 80% of the stingless bee species in Malaysia [3].

Spermatophyta was the most abundant phylum found in this study, followed by Magnoliophyta. Spermatophyta is a plant group that produces seeds [36], while Magnoliophyta produces flowers. Both phyla are very active in producing a lot of pollen. As seed producers, members of Spermatophyta normally depend on pollinators, almost 80% of which are bee species [37,38]. Magnoliophyta, likewise, constantly produces flowers which attract the bees to collect pollen and nectar [39]. However, some of the Magnoliophyta species have a specifically pointed corolla structure to enable the bees to enter the flowers and collect the nectar. In such a situation, the bees will sometimes bore a hole in the corolla to steal the nectar without collecting the pollen from the flowers [40]. Many studies on the food source of stingless bees confirmed that Magnoliophyta and Spermatophyta were the two phyla most frequently visited by stingless bees [41].

Based on the analyzed data, we found that nectar from all the six phyla collected by *H. itama* were represented in the samples from Sarawak Beladin, in which Magnoliophyta was the most common at 78.7% (Table 4). However, the number of OTUs, which was only 39, was not as high as in the other localities (Table 2). Beladin is in the Betong division in Sarawak, where Meludam National Park is located. This national park has its own uniqueness due to some of the plant species recorded here that are poorly recognized [42], and most probably consist of endemic species. The national park still has 3.29% undisturbed area [43] and is identified as a refuge for Sarawak’s most endemic and rarest species [44]. Beladin, Sarawak also presented a very low percentage of Embryophyta (1.19%), but none of the species in this phylum could be found in other localities in Peninsular Malaysia and Borneo. 

The family composition of our study results was dominated by Euphorbiaceae, Fabaceae, Fagaceae, and Cucurbitaceae (Figure 3). Normally, the stingless bees would forage, for their diet, for three types of honey sources, namely, nectar from flowers for nectar honey, nectar from insects for honeydew honey, and nectar from leaf shoots for extrafloral honey [45,46]. The Euphorbiaceae family produces unlimited extrafloral honey, and some of the species are associated with ants [47]. In Malaysia, the stingless bees harvest pollen from the flowers of *Macarangga* sp. and *Hevea* sp. to produce extrafloral honey. Studies by Bahri (2018) [48] highlighted that stingless bees were domesticated under rubber trees to increase yield and production. Although the rubber tree does not produce pollen constantly and only once a year normally, our data have revealed that *Hevea* sp. DNA was found inside the stingless bee bodies, indicating that besides the pollen, DNA could also be derived from other plant parts, as revealed by using metabarcoding. 

From our study results, it can be concluded that in some situations, stingless bees will forage for nectar without collecting pollen. Referring to Jaapar et al. (2018) [49] and Asma et al. (2019) [50], stingless bees collect pollen in the morning from 9 a.m. to 11 a.m., and then again from 3 p.m. to 4 p.m. The worker bees also normally collect resin to produce extrafloral honey before returning to the hive [49,51]. Interestingly, members of Fabaceae, Fagaceae, and Cucurbitaceae normally produce yellowish-colored flowers that are preferably collected by the stingless bees [1], and Fabaceae has been identified as the most frequently visited family for collecting pollen by the stingless bees [52]. Thus, although Fagaceae had been determined as the dominant family in the mixed orchards or secondary forests wherein the meliponiculture activities were sited [53], *H. itama*, however, preferably selected plant species belonging to Fabaceae in their foraging behavior.

Based on the heat map (Figure 4), four species, namely, *Garcinia oblongifolia, Mallotus pellatus, Pinus squamata*, and *Phoebe puwenensis*, were found in all the localities to be foraged by *H. itama* [52,54,55]. According to Azuan et al. (2008) [56], Perak, Perlis, and Kedah showed high densities of *Garcinia oblongifolia* communities. *Garcinia* is a genus that produces fruits and resin for stingless bees, from which the nectar and resins are collected by the bees during the flowering season [57]. We estimated that the DNA of *Garcinia oblongifolia* and *Pinus squamata* was sourced from the plant resins, collected by the bees when foraging for structural materials for their hives [12]. Meanwhile, *H. itama* was attracted to both *Mallatus pellatus* and *Phobe puwenensis* due to the abundance of their nectar and pollen.

According to Nagamitsu et al. (1999) [4], several plants species from the ground to the canopy tops are foraged by stingless bees in their quest for nectar and pollen, which, consequently, would result in their pollination of the flowers. The stingless bee has also been identified as an important pollinator of dipterocarp species in the forests of Malaysia [58], as well as in agricultural areas [50]. However, a very good management strategy is required for the domestication process of stingless bees, especially in mono-cropping systems, as these insects play a major role as the pollinating agent for crops as well as for honey production. 

Based on these findings, the data generated in this study can be utilized in many broad-scale applications. For example, DNA of the durian species *Durio zibethinus* has been detected in the gut of *H. itama,* even though it has not been recognized as a highly abundant species based on the metabarcode data. Therefore, there is a high potential of utilizing *H. itama* as an alternative pollinator to be domesticated in large acreage durian orchards in Malaysia and other tropical countries and is thus highly recommended. A study by Wayo et al. (2018) [59] has confirmed that stingless bee species are also pollinators of durian, besides the well-known bat species. The dual role of stingless bees in pollination and honey production can be fully exploited by farmers and local communities to generate more income. Thus, our study data and findings can be utilized in many areas of application to support our socio-economic activities in the near future. Likewise, in sericulture, the domestication process of stingless bees can be successfully developed under mono-cropping systems and in netted structures by planting all the possible plant species foraged by these bees to produce higher value and better quality honey as well as improved brood production as a promising way forward.

## 5. Conclusion

The taxonomic information (phylum, family, genus, species, OTU, diversity) on the plant species foraged by *H. itama* has been presented in this study through the metabarcoding analysis obtained from the DNA of adult *H. itama*. The fast-growing plant species under the phyla Spermatophyta and Magnoliophyta, such as *Momordica charantia, Melastoma* sp., and *Cucumis sativa,* which produce abundant flowers and nectar, are recommended to be planted in mono-cropping systems for an effective and successful domestication process within the netted structure. Therefore, the food plant species obtained from the metabarcoding analysis in this study are considered as key species in the domestication process of *H. itama* for high-quality honey production and for greater success in the meliponiculture industry.

## Figures and Tables

**Figure 1 insects-12-00205-f001:**
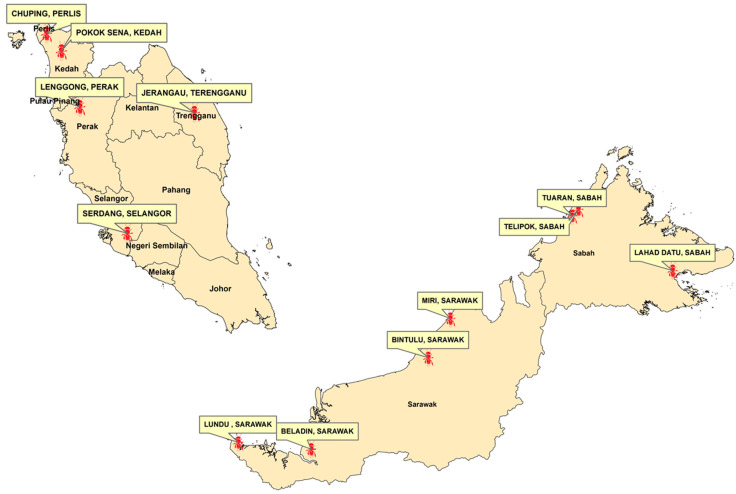
Collection sites of *Heterotrigona itama* samples throughout Malaysia.

**Figure 2 insects-12-00205-f002:**
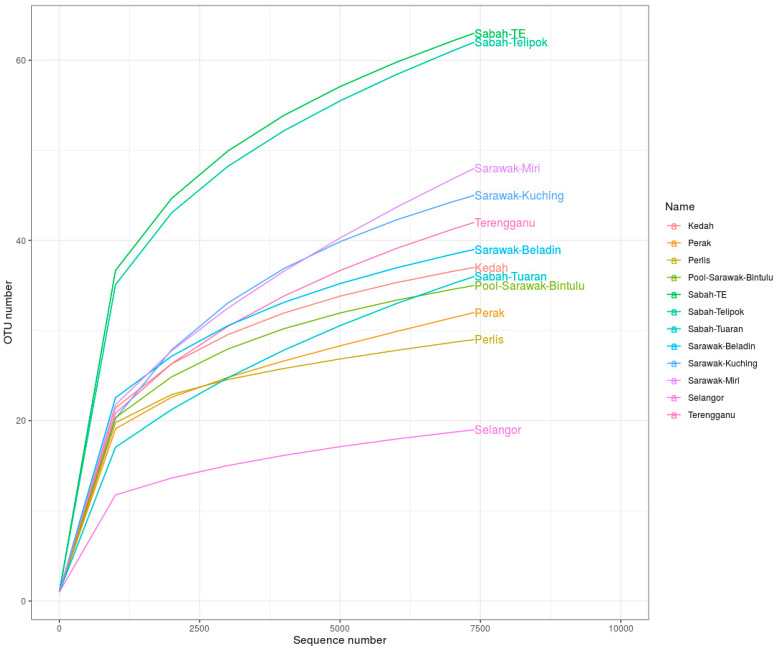
The rarefaction curve of the trnL gene sequence for different localities of *H. itama* calculated for operational taxonomic units (OTUs) at 97% similarity.

**Figure 3 insects-12-00205-f003:**
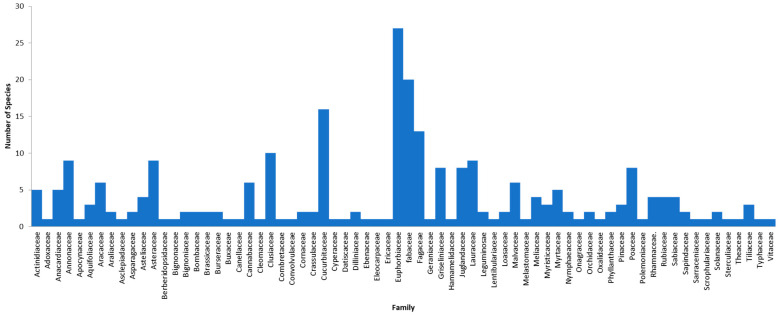
Families of plant species based on trnL result.

**Figure 4 insects-12-00205-f004:**
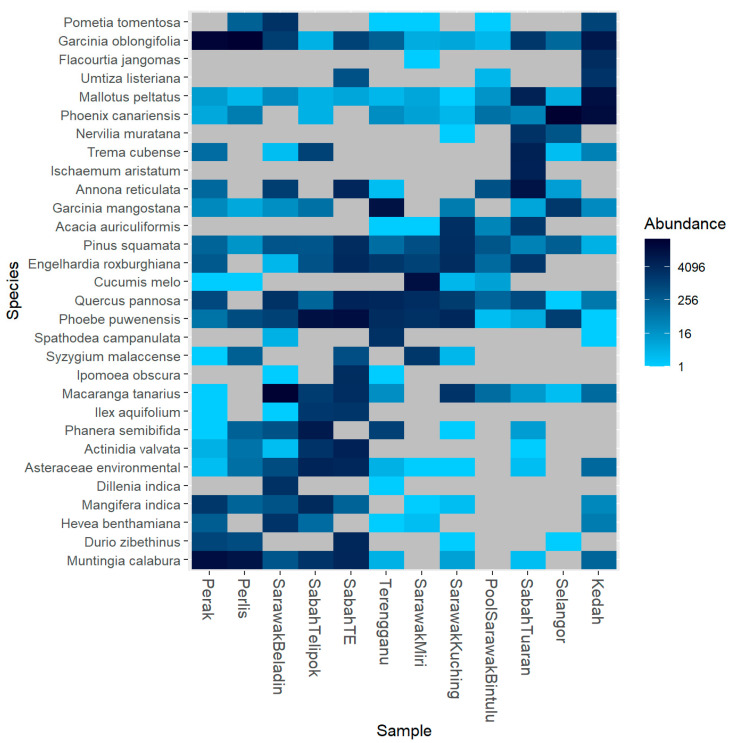
Heatmap at species level. The 30 most abundant genera were used in hierarchical clustering. A darker color indicates a more dominant genus.

**Figure 5 insects-12-00205-f005:**
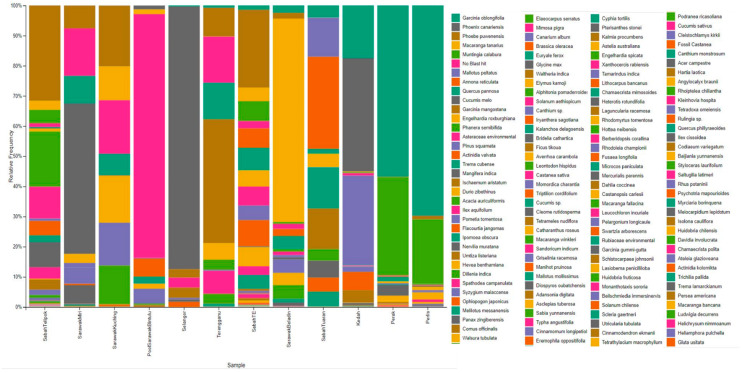
Relative frequency of plant species from different localities.

**Figure 6 insects-12-00205-f006:**
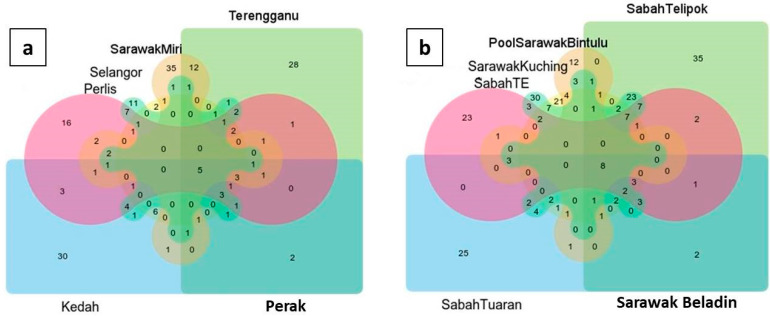
Venn diagrams summarizing the number of shared plant species among different locations. (**a**) Shared plant species between Sarawak and Peninsular Malaysia (Terengganu, Kedah, Perak, Perlis, Selangor). (**b**) Shared plant species between Sabah (Sabah Telipok, Sabah TE, and Sabah Tuaran) and Sarawak (Sarawak Beladin, Pool Sarawak Bintulu, and Sarawak Kuching).

**Figure 7 insects-12-00205-f007:**
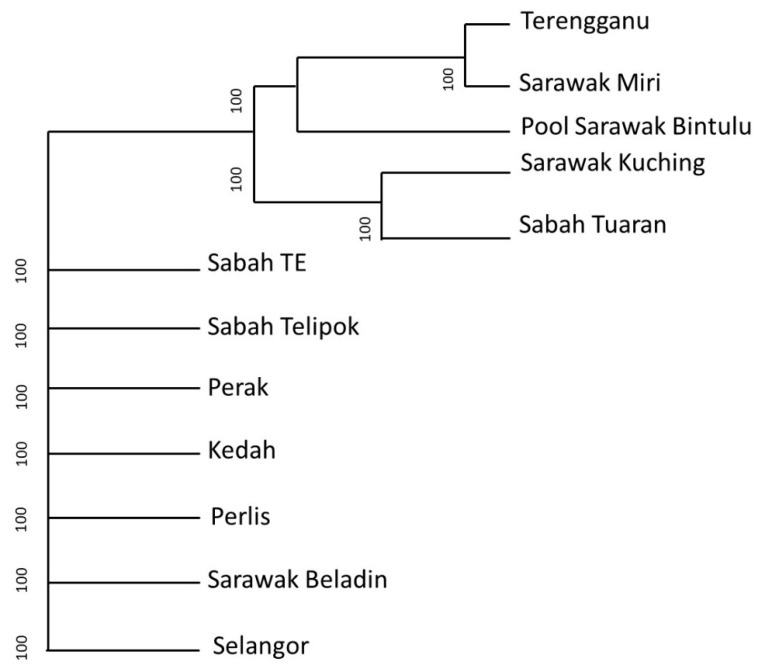
Phylogenetic tree dendrogram based on trnL using Bray–Curtis distance (at family level).

**Table 1 insects-12-00205-t001:** List of samples used for metabarcoding analysis.

Sample Code	Location	Grid	Ecosystem
Sabah TE	Borneo. Sabah: Lahad Datu	5.01248, 118.0970218	Crop field adjacent to secondary forest
Terengganu	Peninsular Malaysia. Terengganu: Jerangau	5.0655177, 102.8652274	Crop field adjacent to secondary forest
Sabah Telipok	Borneo. Sabah: Telipok	6.1732126, 116.2984958	Crop field adjacent to hill forest
Perak	Peninsular Malaysia. Perak: Lenggong	5.1662481, 100.8801384	Crop field
Kedah	Peninsular Malaysia. Kedah: Pokok Sena	6.126510, 100.559629	Crop field
Sarawak Kuching	Borneo. Sarawak: Lundu	1.758078, 109.806310	Crop field adjacent to coastal area
Pool Sarawak Bintulu	Borneo. Sarawak: MARDI Bintulu	3.358613, 113.430816	Crop field adjacent to oil palm estate
Perlis	Peninsular Malaysia. Perlis: Chuping	6.427755, 100.304074	Crop field
Sarawak Beladin	Borneo. Sarawak: Beladin	1.6273364, 111.1958089	Crop field
Selangor	Peninsular Malaysia. Selangor: MARDI Serdang	2.990892, 101.702434	Crop field
Sabah Tuaran	Bornoe. Sabah: Tuaran	6.0523621, 116.1788793	Crop field adjacent to river
Sarawak Miri	Borneo. Sarawak: Miri	4.102966, 113.853467	Crop field adjacent to coastal area

**Table 2 insects-12-00205-t002:** The numbers of effective trnL gene sequences. Number of observed OTUs, alpha diversity indices for the plant DNA from 12 localities of *H. itama.*

Localities	Sequences	OTUs	Chao1	Shannon–Weiner	Simpson	Evenness
Sabah TE	70,068	63	83.000	2.732	0.900	23.056
Terengganu	34,210	42	53.375	2.126	0.826	19.769
Sabah Telipok	50,173	62	87.500	2.372	0.841	26.144
Perak	52,216	32	59.500	1.147	0.564	27.901
Kedah	54,937	37	42.600	1.672	0.739	22.126
Sarawak Kuching	20,642	45	54.75	2.023	0.851	22.244
Pool Sarawak Bintulu	7448	35	42.000	1.939	0.794	18.047
Perlis	56,612	29	34.000	1.044	0.476	27.786
Sarawak Beladin	55,714	39	48.000	1.494	0.541	26.098
Selangor	47,532	19	22.330	0.645	0.243	29.470
Sabah Tuaran	44,060	36	53.500	2.130	0.840	16.899
Sarawak Miri	33,786	48	86.500	1.857	0.715	25.844
Total	527,398	487				

**Table 3 insects-12-00205-t003:** Relative abundance at the phylum level of plant communities from different localities.

Phylum	Localities %
Tracheophyta	4.72
Anthophyta	1.97
Magnoliophyta	32.39
Spermatophyta	55.95
Embryophyta	0.13
Unknown	4.85

**Table 4 insects-12-00205-t004:** Relative abundance at the phylum level of plant communities in *Heterotrigona itama* from different localities.

Phylum	Borneo (%)							Peninsular Malaysia (%)				
	Sabah TE	Sabah Telipok	Sabah Tuaran	Sarawak Kuching	Pool Sarawak Bintulu	Sarawak Beladin	Sarawak Miri	Terengganu	Perak	Kedah	Perlis	Selangor
Tracheophyta	6.66	2.81	18	15.7	1.6	0.11	9.39	8.9	1.114	0.9	0.08	0
Anthophyta	6.32	10.56	0.01	0.01	0	1.13	0.01	0.01	0.01	0.3	0.17	0.05
Magnoliophyta	41.92	57.37	38.3	39	4.09	78.7	22.4	32.5	2.9	34.7	5.6	3.6
Spermatophyta	42.7	27.9	43.5	27.48	13.4	17.1	52.3	43.2	95.8	63.2	94.1	93
Embryophyta	0	0	0	0	0	1.16	0	0	0	0	0	0
Unknown	2.3	1.253	0.014	17.7	80.8	1.64	15.7	15.2	0.14	0.7	0.01	3.2

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
