# Peer review of "Metabarcoding in Diet Assessment of Heterotrigona itama Based on trnL Marker towards Domestication Program"

_insects, 2021, doi:10.3390/insects12030205_

Round 1

Reviewer 1 Report

Heterotrigona itama is a highly eusocial solitary bee, which has been domesticated in Malaysia for commercial pollination as well as for honey production.  To propagate new colonies domestically, a fundamental understanding of the biology of the bees including their diet is necessary. Authors here studied the metabarcode data from 12 populations of H. itama throughout Malaysia (Borneo and Peninsular Malaysia) using trnL marker. This is of interest for farmers and beekeepers beyond Malaysia.

In the general well-done study, 70 families of diet plants under 262 species were recorded from all the populations. The present manuscript is principally suitable for publication in Insects but needs considerable editorial and language improvement.

Just to mention some of the inadequacies:

  • I miss some more recent literature on bee domestication in the meliponiculture industry such as the book by Atsalek Rattanawannee and Orawan Duangphakdee: Southeast Asian Meliponiculture for Sustainable Livelihood. DOI: http://dx.doi.org/10.5772/intechopen.90344
  • species names should not be used with “the”
  • grammatical errors to be corrected such as in lines 49/50: “To date 50,000 colonies of H. itama have been reared for honey production in Malaysia and this number is increasing yearly”, and many more!
  • Table 1: the column “Species” has to be deleted; only one species was used
  • line 114 and others: insert a blank between numbers and units
  • line 129 and others: it is not necessary to describe the type of pipette used!
  • line 163: Shannon-Wiener index is the correct name
  • legend to Fig. 3 and others: 16S rRNA is the correct name
  • line 183: five in lowercase letters
  • line 213: insert commas between species names
  • Fig. 4 does not show relative abundances at the “family” level and data are redundant to those in Table 4
  • the last paragraph of the Discussion sounds like a part of a reviewer report
  • References have to be carefully checked for a uniform style of presentation

Author Response

 1. REVIEWER 1: Heterotrigona itama is a highly eusocial solitary bee, which has been domesticated in Malaysia for commercial pollination as well as for honey production. To propagate new colonies domestically, a fundamental understanding of the biology of the bees including their diet is necessary. Authors here studied the metabarcode data from 12 populations of H. itama throughout Malaysia (Borneo and Peninsular Malaysia) using trnL marker. This is of interest for farmers and beekeepers beyond Malaysia. In the general well-done study, 70 families of diet plants under 262 species were recorded from all the populations. The present manuscript is principally suitable for publication in Insects but needs considerable editorial and language improvement.

AUTHORS: Thank you for your review and comments. On behalf of the authors, I would like to thank you for your time and valuable comments. According to your comment, English needs moderate revision. For your information, the manuscript has been proofread for appropriate English language usage, grammar, punctuation and spelling by an experienced English-speaking editor.

Just to mention some of the inadequacies:

2. REVIEWER 1: I miss some more recent literature on bee domestication in the meliponiculture industry such as the book by Atsalek Rattanawannee and Orawan Duangphakdee: Southeast Asian Meliponiculture for Sustainable Livelihood. DOI: http://dx.doi.org/10.5772/intechopen.90344.

AUTHORS: The book has been cited in the text (Line 47-48) Due to their abundancy, the H. itama is domesticated in Malaysia meliponiculture industry (Jaapar and Jajuli 2015), as well as in the neighbourhood countries (Rattanawannee and Duangphakdee 2019).

3. REVIEWER 1: Species names should not be used with “the”.

AUTHORS: All has been revised throughout manuscript.

4. REVIEWER 1: Grammatical errors to be corrected such as in lines 49/50: “To date 50,000 colonies of H. itama have been reared for honey production in Malaysia and this number is increasing yearly”, and many more!

AUTHORS: The sentence has been edited and several grammatical errors also corrected throughout the manuscript by the proof-reader.
To date, there are 50,000 colonies of H. itama have been reared for honey production in Malaysia and this number is increasing yearly. (Line 48-49).

5. REVIEWER 1: Table 1: the column “Species” has to be deleted; only one species was used.

AUTHORS: Column ‘species’ has been deleted.

6. REVIEWER 1: Line 114 and others: insert a blank between numbers and units AUTHORS: Has corrected all places on the manuscript.

7. REVIEWER 1: line 129 and others: it is not necessary to describe the type of pipette used!

AUTHORS: The information (type of pipette used) has been deleted. Line 118-135.

8. REVIEWER 1: Line 163: Shannon-Wiener index is the correct name.

AUTHORS: Has been corrected. Line 163.

9. REVIEWER 1: Legend to Fig. 3 and others: 16S rRNA is the correct name.

AUTHORS: Has been corrected. Line 182.

10. REVIEWER 1: Line 183: five in lowercase letters.

AUTHORS: The mistake has been corrected. Line 183.

11. REVIEWER 1: Line 213: insert commas between species names

AUTHORS: Has been edited.

12. REVIEWER 1: Fig. 4 does not show relative abundances at the “family” level and data are redundant to those in Table 4.

AUTHORS: Figure 4 deleted and only remain the Table 4 only.

13. REVIEWER 1: The last paragraph of the Discussion sounds like a part of a reviewer report. Authors should discuss the results and how they can be interpreted from the perspective of previous studies and of the working hypotheses. The findings and their implications should be discussed in the broadest context possible. Future research directions may also be highlighted.

AUTHORS: A paragraph has added with the information suggested. Line 355-368.
Based on these findings, the generated data can be implemented in many broad-scale applications. For example, durian species, Durio zibethinus has been detected in the gut of H. itama, even though it was not determined as highly abundant species based on the metabarcode data. Therefore, the potential of H. itama as an alternative pollinator to be domesticated in hectares of durian orchard in Malaysia and other tropical countries is highly suggested and recommended. A study by Wayo et al. (2018) [54] has confirmed that stingless bee species have become a pollinator of durian, besides of bat species. Instead of their function as pollinator, the honey production may be generated by the farmers as their side income. Thus, detail implications based on the findings can be generated in many more significant economic applications in the near future. Furthermore, all the possible plant species to be planted in the mono-cropping system and in netted structure has successfully achieved towards domestication process. Consequently, the value and honey quality, as well as brood production of H. itama are become the way forwards from this successful study.

Wayo, K., Phankaew, C., Stewart, A., & Bumrungsri, S. (2018). Bees are supplementary pollinators of self-compatible chiropterophilous durian. Journal of Tropical Ecology, 34(1), 41-52. doi:10.1017/S0266467418000019

14. REVIEWER 1: References have to be carefully checked for a uniform style of presentation.

AUTHORS: References have been rechecked

Reviewer 2 Report

This is an interesting study on metabarcoding of diet of stingless bee. The results the authors received have important implications for further exploring ecosystem services and identification of Heterotrigona itama host plants. Overall the manuscript is very well-written and organized; all the figures and tables are well prepared. The authors provided the detailed introduction to the topic and its discussion; the experiments are well designed, thoroughly conducted and well described. 

I have a few comments for the authors to consider (see below):

the title. Should it be "program" there?
Line 70. Do you mean DNA barcoding? ("the molecular application" sounds confusing)
Line 93. Table 1. Please replace "Agriculture" with either "managed ecosystem" or "managed landscape", or "crop field", etc. depending on the type of ecosystem the samples were collected from 

Author Response

REVIEWER 2: This is an interesting study on metabarcoding of diet of stingless bee. The results the authors received have important implications for further exploring ecosystem services and identification of Heterotrigona itama host plants. Overall the manuscript is very well-written and organized; all the figures and tables are well prepared. The authors provided the detailed introduction to the topic and its discussion; the experiments are well designed, thoroughly conducted and well described.
AUTHORS: Thank you very much for your positive comments. On behalf of the authors, we would like to thank you for the valuable comments.

I have a few comments for the authors to consider (see below):

2. REVIEWER 2. The title. Should it be "program" there?

AUTHORS: Depending on journal. Journal has highlighted either British or American English may be used under the author’s guideline.

3. REVIEWER 2: Line 70. Do you mean DNA barcoding? ("the molecular application" sounds confusing).

AUTHORS: Yes, we would like to highlight DNA barcoding. The reference has been changed to to Xiong et al. (2018) [16]. (Line 70-71).

The DNA barcoding application becomes the latest approach or tool in biotechnology, which showing implifications on human health and marine sustainability [16]. The application of the DNA molecule also has solved many problems and issues related to the meliponiculture industry worldwide, such as to access the honey compound in several honey varieties [17] …..,

4. REVIEWER 2: Line 93. Table 1. Please replace "Agriculture" with either "managed ecosystem" or "managed landscape", or "crop field", etc. depending on the type of ecosystem the samples were collected from

AUTHORS: The agriculture field changed to crop field. Information in the table 1 that related to agriculture also edited (Line 93-94).